# Runtime Analysis of Area-Efficient Uniform RO-PUF for Uniqueness and Reliability Balancing

**Zulfikar Zulfikar** [1,2,*], **Norhayati Soin** [2,3], **Sharifah Fatmadiana Wan Muhamad Hatta** [2,3] **and Mohamad Sofian Abu Talip** [2]

1. Department of Electrical and Computer Engineering, Universitas Syiah Kuala, Jl. Teuku Nyak Arief, Darussalam, Banda Aceh 23111, Indonesia
2. Department of Electrical Engineering, Faculty of Engineering, Universiti Malaya, Kuala Lumpur 50603, Malaysia; norhayatisoin@um.edu.my (N.S.); sh_fatmadiana@um.edu.my (S.F.W.M.H.); sofian_abutalip@um.edu.my (M.S.A.T.)
3. Center of Printable Electronics, Universiti Malaya, Kuala Lumpur 50603, Malaysia
* Correspondence: zulfikarsafrina@unsyiah.ac.id

**Abstract:** The main issue of ring oscillator physical unclonable functions (RO-PUF) is the existence of unstable ROs in response to environmental variations. The RO pairs with close frequency differences tend to contribute bit flips, reducing the reliability. Research on improving reliability has been carried out over the years. However, it has led to other issues, such as decreasing the uniqueness and increasing the area utilized. Therefore, this paper proposes a uniform RO-PUF, requiring a smaller area than a conventional design, aiming to balance reliability and uniqueness. We analyzed RO runtimes to increase reliability. In general, our method (uniqueness = 47.48%, reliability = 99.16%) performs better than previously proposed methods for a similar platform (Altera), and the reliability is as good as the latest methods using the same IC technology (28 nm). Moreover, the reliability is higher than that of RO-PUF with challenge and response pair (CRP) enhancements. The evaluation was performed in longer runtimes, where the pulses produced by ROs exceeded the counter capacity. This work recommends choosing ranges of the runtime of RO for better performance. For the 11-stage ROs, the range should be 1.598–4.30 ms, or 6.12–8.61 ms, or 12.24–12.91 ms. Meanwhile, for the 20-stage, the range should be 2.717–8.37 ms, or 10.97–16.74 ms, or 21.93–25.10 ms.

**Keywords:** area-efficient; FPGA; uniform RO-PUF; routing density; runtime

## 1. Introduction

The first ring oscillator PUF (RO-PUF) model was proposed by Gassend et al. [1,2]. The authors adopted the Arbiter PUF for extracting the delay of the switch box route. The oscillation was obtained by transferring the delay of the circuit via negative feedback. For enabling and looping the oscillations, an AND gate was added to the first stage. However, their idea had minor drawbacks. The design is susceptive to model attacks because the RO is based upon the identical chain as the Arbiter PUF. The circuit requires a detailed design to be incorporated for implementation purposes. Moreover, the produced response is a value that could not be directly used in the following blocks. Another alternative of RO-PUF was introduced in 2007 [3]. This design can produce bitwise naturally. The design accommodates a fixed number of identical array-realized ROs. The design involves two counters for comparing the two chains at one time. A sequence response bit was successful in response to several different challenges.

Since then, many researchers have participated in developing RO-PUF. Chi and Gang proposed a temperature-aware cooperative (TAC) technique by converting RO pairs that produce unreliable bits to be reliable bits [4]. Merli et al. evaluated the RO's frequency on the Xilinx field programmable gate arrays (FPGA) to improve its performance [5]. The authors evaluated multi-stage ROs over various runtimes. Kodytek et al. then proposed

ROs that could provide more output bits from each RO chain pair and are not dependent on the symmetry, besides having no restriction on the RO placement [6,7]. Then, the first aging-resistant RO-PUF design was proposed in 2016 by Rahman et al. [8]. It was designed to reduce sensitivity to negative bias temperature instability (NBTI) and hot carrier injection (HCI) stress. The ideas were realized by replacing the error correction scheme with extra ROs. Hence, RO frequency degrades at a much slower rate and results in fewer bit flips in PUF over time.

Other investigations showed that the RO frequency is interference by other circuits. Gag et al. reported on the crosstalk effects in FPGA and provided a demonstration of digital conversion for measuring the coupling capacitance's impact when analyzing interconnections [9]. The electronic design automation (EDA) tools often automatically connect the path inside the FPGA chip. The connection might cause an imbalance in routing in some locations. Ikeda et al. discussed this issue, which affects RO performance, especially by reducing uniqueness [10]. The issue was later discussed further by Giechaskiel et al. by showing that the pulses that result in ROs are interference by other signals [11]. A similar issue was also investigated further in 2019 [12].

The main issue of RO-PUF is the existence of unstable ROs in response to environmental variations. The RO pairs with close frequency differences tend to contribute bit flips, reducing the reliability. Research on reducing the issue has been carried over the years. However, it has led to other issues, such as decreasing the uniqueness and increasing the area utilized.

Therefore, an RO-PUF that balances area utilization, reliability, and uniqueness is required. This work proposes a uniform RO-PUF which requires a lesser area compared to conventional designs. The ROs are connected directly to counters to enhance the challenge and response pair (CRP). Routing density among ROs is designed to be uniform to increase uniqueness. With the uniform RO-PUF, this work investigated the RO runtimes to reduce bit flip further. The evaluation was done with longer runtimes where the pulses produced by ROs have exceeded the counter capacity.

## 2. The Proposed RO-PUF

In most previous studies, the performance was estimated based on the frequency of pulses generated by the RO [3,5,7,8,13–19]. The frequency of RO pulses was measured using an oscilloscope or a logic analyzer. Upon CRP generation, most methods employ two counters, which may cause locking phenomena [14,20,21]. The phenomena may reduce the uniqueness of the RO-PUF. Later, RO-PUF proposed by Maiti et al. [22] and Delavar et al. [23] were actualized with only one counter. However, both approaches resulted in inequality of the routing distance from ROs to counter. This work proposes a uniform RO-PUF, a structure where routing hotspots among ROs are as narrow as possible. This work refers to the close routing hotspots among ROs as "uniform." With the uniform design, all ROs would expose similar noises. Hence, a purer physical variation of individual RO can be obtained [24]. Unlike the conventional design where between RO and the counter exists another circuit, the proposed RO-PUF connects RO directly to the counter.

### 2.1. RO-PUF Circuit

The circuit of the proposed RO-PUF is shown in Figure 1. It consists of four subcircuits: a controller, ROs, counters, and CRP generation. These circuits are placed in separate blocks to adjust the routing density of ROs to become uniform. The controller is used to manage RO runtimes. CRP generation is used to generate the response. Input *Enable* is used to trigger the activation of an RO. Each RO is connected directly to a particular counter. Therefore, the proposed RO-PUF gains the benefits of:

- High RO utilization: a higher response bit can be achieved, as CRP generation is independent.

- Routing equality: the distance between each RO and the respective counter is equal. Hence, there will be no locking phenomena or jitter noise.

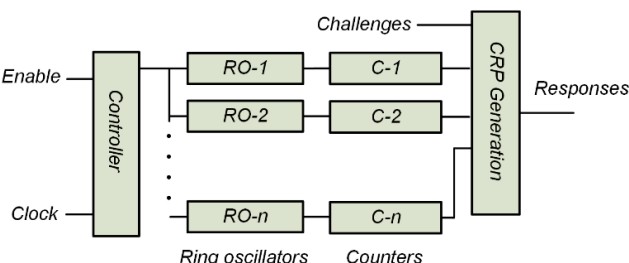

**Figure 1.** The proposed uniform RO-PUF.

### 2.2. Ring Oscillator Realization

According to previous research, three-stage RO is less stable [14,25,26]. The design resulted in the worst uniqueness [14]; the difference in frequency between ROs is 80 MHz, which is relatively wide [25]; and the resulting waveforms are challenging to detect by the counter [26]. Therefore, this study examines 5-stage, 11-stage, and 20-stage ROs. The highest number of stages being 20 is due to the size of logic array blocks (LAB). In the remainder of the paper, 5-stage, 11-stage, and 20-stage ROs are abbreviated as RO5, RO11, and RO20, respectively.

Based on several previous studies [14,20,27,28], specific methods are needed to realize ROs on the Altera FPGA chips. The LAB of Cyclone V consists of ten adaptive logic modules (ALM). There are two LUTs and four flip-flops in each ALM. Hence, to realize RO5, RO11, and RO20, it takes three ALMs, six ALMs, and ten ALMs, respectively. Theoretically, 5-stage RO was composed using a 2-input NAND gate and four NOT gates. However, the straightforward implementation of the circuit was not permissible. Therefore, hard macro-calibration was required [24,29,30] for equal functionality. Moreover, the technique is more flexible regarding the use of even or odd numbers of buffers. Figure 2 shows the realized RO5 inside the Cyclone-V built using a modified NAND and four buffers.

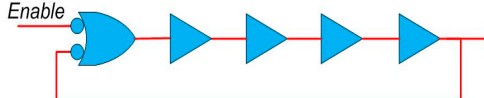

**Figure 2.** RO5 implemented inside the Cyclone V-chip.

### 2.3. RO-PUF Placement

Maiti et al. [31] were the first to investigate the RO placement in the Xilinx FPGA and observe the frequency difference. Later, Feiten et al. explored several positions of the Altera chip; they found that the RO at the corner of the chip had lower reliability [14]. They also found that the distance of the controller's position concerning the RO location did not affect performance. However, we placed RO-PUF at the corner of the chip, as shown in Figure 3 by the red box, to avoid further interference by the primary circuit. The placement of all ROs in one location was intended to uniformize the routing procedure explained in Section 3.1. Figure 3a shows the position of the circuit in the 5CSEMA5F31C6 chip (DE1-SoC board), which was the bottom right corner. Figure 3b shows the circuit position in the 5CSEMA4U23C6 chip (DE0-Nano board), which was the upper-left corner where the area could still be synchronized with one clock.

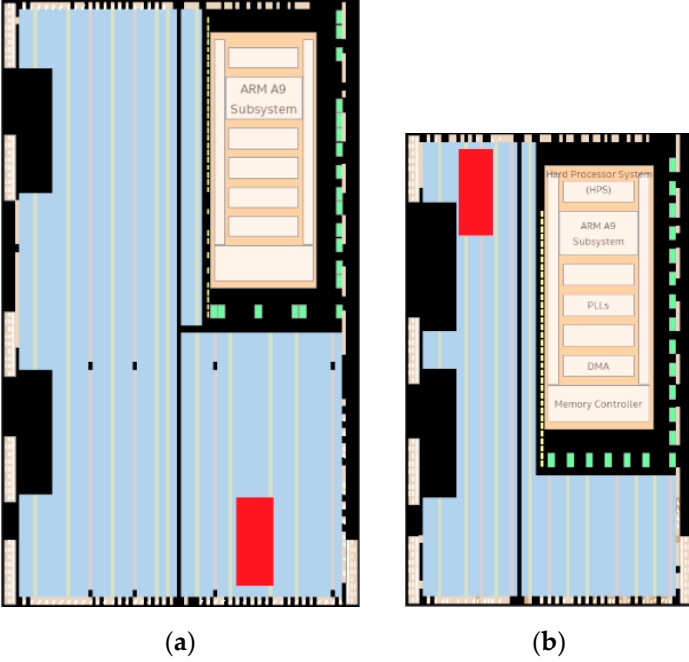

(**a**)                    (**b**)

**Figure 3.** Red boxes signify RO-PUF of (**a**) 5CSEMA5F31C6 (DE1-SoC) and (**b**) 5CSEMA4U23C6 (DE0-Nano).

### 2.4. RO Patterns

Most researchers arrange gates to create the RO in the same way. The NAND gate is at the top of the configurable logic block (CLB) or LAB, and the output gate is at the bottom. We arranged gates in LABs using three arrangements (Pattern-1, Pattern-2, and Pattern-3) for RO5 and RO11, as shown in Figure 4. However, there are only two possible arrangements for RO20, which are Pattern-1 and Pattern-2. In the arrangement of Pattern-1, the NAND gate is created at the top, and the output gate is at the bottom. The arrangement of Pattern-2 is the opposite to Pattern-1; the NAND and output gates are positioned at the bottom and top of the LAB, respectively. Finally, Pattern-3 follows the sequence of Pattern-1, but the RO is shifted to the middle of the LAB.

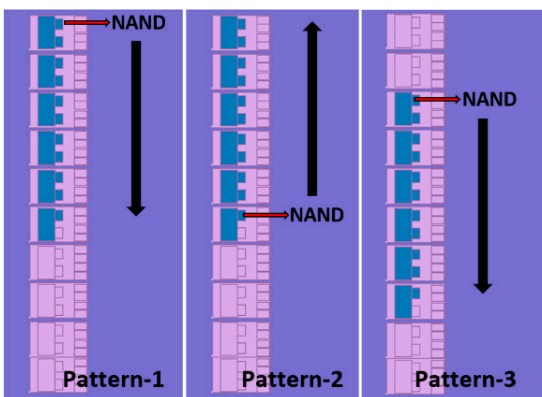

**Figure 4.** Arrangements of gates inside a LAB of Cyclone V-chip for 11-stage RO.

### 2.5. Routing Density

Quartus assigned efficient routing allows a design to be realized for lessening space and delay. The problem of routing is architecture-dependent for the vaious FPGA technologies available in the market. There are the local, direct-link, row, and column interconnects in Cyclone V, as viewed in Figure 5 [32]. The connection or route of logic inside the LAB is employed via local interconnects; the direct link may connect the adjacent block's logic.

Meanwhile, the row and column interconnects are used for connecting blocks. The row connections are R3, R6, and R14, connecting to 3, 6, and 14 adjacent horizontal LABs. The column connection of C2, C4, and C12 connects up to 2, 4, and 12 LABs vertically. Half of the interconnects are utilized to establish connections to one direction in every routing type. The term "routing hotspots" is used when estimating FPGA chips' routing density. It refers to the maximum percentage of interconnecting distribution in any direction in a particular row or column [33]. Consequently, Quartus estimates routing hotspots at each LAB are based on the maximum total wire employed to any direction in C2, C4, C12, R3, R6, or R14.

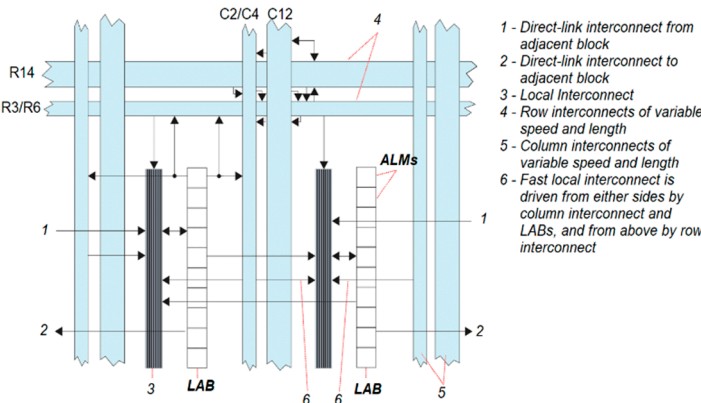

**Figure 5.** Cyclone V routing architecture.

### 2.6. RO Runtimes

As the runtime increases, the RO's resulting pulses may exceed the counter capacity, leading to inaccurate results [5] or incorrect counter value reading [7]. In order to improve the accuracy due to pulses exceeding the counter capacity, we estimated the appropriate runtimes by considering pulse deviations due to extreme environmental changes. Therefore, to estimate the RO-PUF performance at longer runtimes, this study activated RO to generate pulses more than three times the counter capacity. A 20-bit counter is needed to avoid jitter noise [14] or locking phenomena [20,21] due to unequal route distances from RO to counter.

A pre-test was performed to estimate and implement RO at various runtimes. The idea was to pick the proper runtime to avoid the pulses falling near the counting edges (gray areas), as shown in Figure 6. The figure shows the general plot of predicted pulses of ROs concerning runtimes. Cc, 2Cc, and 3Cc refer to the counter capacities of one, two, and three cycles. The generated pulses should be produced in the green areas. Any pulses falling outside the green areas would result in bit flip responses (after environmental changes). For instance, whenever the runtime is between $t_{\_H1}$ and $t_{\_L2}$, the pulse of some ROs might cross the counter capacity, resulting in the flipping of some response bits, which constitutes an unreliable response.

This study assumed that the lowest possible operating temperature is 0 °C, and the highest one is 85 °C. In Figure 6, $P_s(t)$ represents the pulses recorded at the highest temperature ($T_2$). $P'_s(t)$ shows the pulses predicted for 85 °C. $P_f(t)$ is a line to represent the highest pulses recorded at the minimum temperature ($T_1$), and $P'_f(t)$ shows the pulses predicted for 0 °C. The pulses were estimated by comparing the pulses recorded at low and high temperatures of the respective RO chain. $t_{\_H1}$, $t_{\_H2}$, and $t_{\_H3}$ are runtimes when the $P'_f(t)$ reached Cc, 2Cc, and 3Cc, respectively. Hence, $\Delta P_{\_H1}$, $\Delta P_{\_H2}$, and $\Delta P_{\_H3}$ are the pulse differences between Cc, 2Cc, and 3Cc against $P_f(t)$ at $t_{\_H1}$, $t_{\_H2}$, and $t_{\_H3}$. Meanwhile, $t_{\_L2}$, and $t_{\_L3}$ are the runtimes where $P'_s(t)$ reached Cc and 2Cc, respectively. Therefore, $\Delta P_{\_L2}$, and $\Delta P_{\_L3}$ are the pulse differences between Cc and 2Cc against $P_s(t)$.

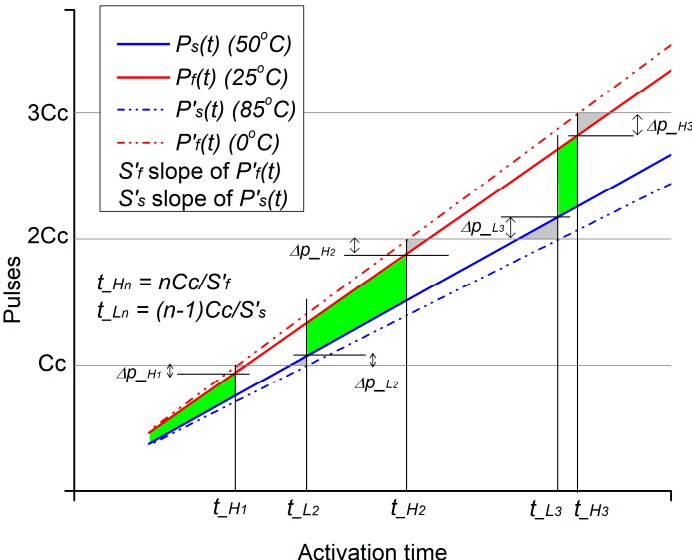

**Figure 6.** Prediction of RO pulses in longer runtimes. ($\Delta P_{\_Hn}$ is the pulse difference between $nCc$ and $P_f(t)$ at $t_{\_Hn}$; $\Delta P_{\_Ln}$ is the pulse difference between $(n-1)Cc$ and $P_s(t)$ at $t_{\_Ln}$. Cc denotes counting capacity.)

First, $S_f$ and $S_s$, which are the slopes of $P_f(t)$ and $P_s(t)$, were determined based on recorded data. Then, the rates of pulse change at the temperature of the fastest RO ($r_f$) and the slowest ($r_s$) can be calculated using Equations (1) and (2).

$$r_f = \left( \frac{P_f(t)\_T_1 - P_f(t)\_T_2}{T_2 - T_1} \right), \tag{1}$$

$$r_s = \left( \frac{P_s(t)\_T_2 - P_s(t)\_T_1}{T_2 - T_1} \right), \tag{2}$$

After that, the pulses of the fastest and slowest ROs in 0 °C ($P'_f(t)$) and 85 °C ($P'_s(t)$) can be determined using Equations (3) and (4).

$$P'_f(t) = P_f(t) + r_f T_1, \tag{3}$$

$$P'_s(t) = P_s(t) - r_s(85 - T_2), \tag{4}$$

Then, $S'_f$ and $S'_s$, slopes of $P'_f(t)$ and $P'_s(t)$ can be determined. The next step is to estimate the runtimes ($t_{\_Hn}$ and $t_{\_Ln}$) at the edges of counter capacity. A further increase in runtimes leads to a narrower green area. When $t_{\_Ln} \geq t_{\_Hn}$, there will be no visible green area. The numbers of pulse differences at particular runtimes between a low and a high temperature and the lowest and highest temperatures with respect to counter capacity can be estimated using Equations (5) and (6). The percentages reflect the maximum possible number of pulse changes at the extreme temperatures.

$$\Delta P_{\_Hn} = 100 - \left( 1 - \frac{t_{Hn} Sf}{nCc} \right), \; where \; n = 1, 2, 3, 4, \tag{5}$$

$$\Delta P_{\_Ln} = 100 - \left( \frac{t_{Ln} Ss}{Cc(n-1)} - 1 \right), \; where \; n = 2, 3, 4, \tag{6}$$

## 3. Experimental Setup

We realized the RO-PUF design explained in Section 2, as explained here. This section also covers uniformizing routing hotspots, the LUT configuration, and the means of measuring the pulses. The implementations were performed using Quartus Prime

Standard. The project, which contained all designs, is compiled until the Fitter (Place and Route) stage. After that, the routing hotspot was generated using Chip Planner. Whenever significant differences existed in routing hotspots among ROs, some logic gates associated with the respective positions were relocated before recompiling the project. Then, the steps were repeated until the differences in routing hotspots among ROs were as narrow as possible. After that, the project was continued to the next step (Assembler) to generate the program file (SOF). The SOF file, containing all the details of data targeted to a specific chip, may be downloaded onto the FPGA chip. The experimental conditions of this work is summarized in Table 1.

**Table 1.** Experimental conditions of the proposed RO-PUF.

| FPGA Technology | Intel Cyclone V (28 nm) |
|---|---|
| Board Chip no. | Four DE1-SoC (5CSEMA5F31C6), six DE0-Nano (5CSEMA4U23C6) |
| RO stage | 5, 11, 20 (e.g., RO11-6 denote RO no. 6 of an 11-stage RO) |
| RO runtimes | 1–8 ms (RO5), 2–16 ms (RO11), 4–32 ms (RO20) |

### 3.1. Uniformizing Routing Hotspots

In new technology, the majority delay in FPGA is due to routing. Therefore, to eliminate physical variations, the routing delay of RO chains should be equal. Feiten et al. have addressed this issue, stating that to eliminate an RO design's physical variations, all ROs should be assigned similar LUT input configurations [29]. However, the idea maintains the internal LAB route only. Therefore, we also considered uniform routing designs suggested in [24] to optimize the routing density outside LAB; details of the LUT inputs can be managed when necessary [29].

Noise influences the performance of the ring oscillators. In non-uniform routing, some ROs may be exposed to more noise than others. Hence, those ROs may generate fewer pulses compared to others. Therefore, the differences in routing, i.e., routing hotspots (Quartus), must be minimized. After the Fitter stage, routing hotspots of all ROs were varied over a range of more than 50%. All subcircuits shown in Figure 1 were located in specific regions. The following steps attempt to uniformize routing hotspots among ROs of the RO-PUF design shown in Figure 1.

- Rearrange the logic fan-out of input pins. Logic gates corresponding to the *Clock* and *Enable* inside the controller are rearranged.
- Rearrange the logic fan-in of output pins. Logic gates corresponding to display *responses* inside CRP Generation are rearranged.
- Distribute logic fan-in of ROs uniformly. All logic gates inside the Controller connected directly to ROs are distributed uniformly in the entire region.
- Distribute logic fan-out of counters uniformly. All logic gates in the CRP Generation region connected directly to counters are rearranged to be uniformly distributed in the entire region.
- Rearrange other logics in the CRP Generation region. Several logics inside this region do not have to be uniformly distributed. However, they are relocated based on the routing hotspots in the respective ROs.

After applying the uniformizing procedures, this study reached a small range of routing hotspots among ROs. The maximum hotspot difference was observed to be 23%. Table 2 lists the routing hotspots of all patterns in both DE1-SoC and DE1-Nano boards.

**Table 2.** RO routing hotspots of all patterns in DE1-SoC and DE0-Nano.

| | RO5 | | | RO11 | | | RO20 | |
| | Pattern-1 | Pattern-2 | Pattern-3 | Pattern-1 | Pattern-2 | Pattern-3 | Pattern-1 | Pattern-2 |
|---|---|---|---|---|---|---|---|---|
| | | | | DE1-SoC | | | | |
| Routing hotspots (%) | 21–42 | 35–57 | 28–50 | 35–50 | 42–64 | 35–57 | 42–64 | 35–57 |
| Average (%) | 30.6 | 45.4 | 36.5 | 46.2 | 51 | 43.7 | 52.6 | 44.6 |
| | | | | DE0-Nano | | | | |
| Routing hotspots (%) | 7–22 | 8–28 | 11–21 | 21–35 | 21–35 | 25–37 | 19–42 | 19–42 |
| Average (%) | 16.4 | 18.7 | 15.1 | 27.3 | 25.1 | 29.3 | 26.6 | 28.7 |

### 3.2. LUT Configuration

The gates to form the ROs were created using LUTs inside LAB. Figure 7 shows the two 6-input LUTs inside an ALM of Cyclone V. There is a notable difference between the LAB of Cyclone V and that of the previous models. The LAB of Cyclone II, III, and IV consists of 16 logic elements (LEs). There is only one LUT inside an LE. In comparison, the LAB of Cyclone V consists of 10 ALMs, and each ALM is constructed from two LUTs. Both LUTs inside an ALM function as 6-input LUTs; the previous one is 4-input.

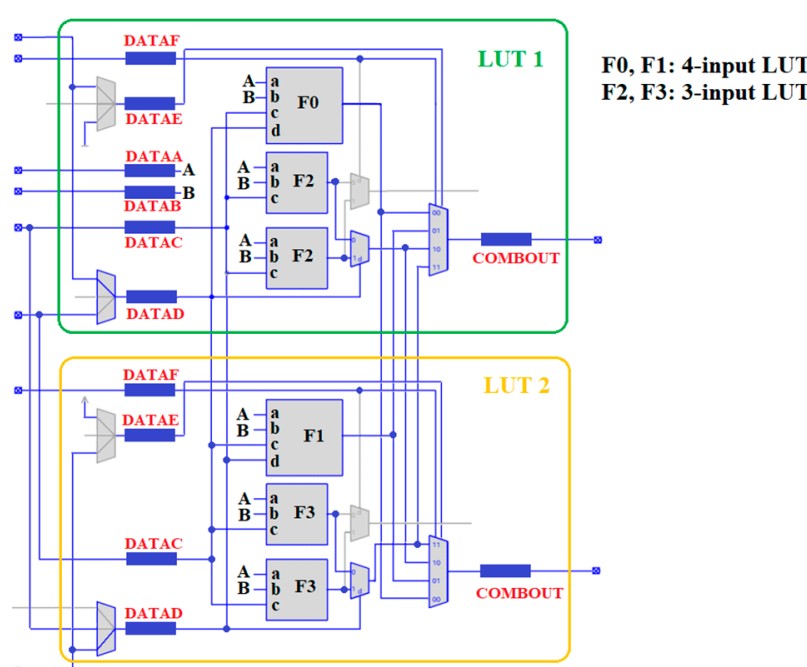

**Figure 7.** Configuration of two LUTs inside an ALM of Cyclone V.

Table 3 lists LUT input configurations concerning the stages and patterns of this study. For instance, Pattern-1 of RO5 inside the DE1-SoC was assigned FB-DCFF, where FB refers to NAND inputs configured via DATA**F** and DATA**B**. The remaining DCFF refers to four buffers' input, as shown in Figure 2, configured via DATA**D**, DATA**C**, DATA**F**, and DATA**F**. Different input configurations can impact the delay of the RO of the respective pattern. For instance, whenever a buffer is configured via input DATA**F**, the buffer's delay is less than other inputs.

**Table 3.** Configuration of LUT inputs of all patterns in DE1-SoC and DE0-Nano.

| Pattern/Stage | | Inputs Configuration | |
| --- | --- | --- | --- |
| | | **DE1-SoC** | **DE0-Nano** |
| Pattern-1 | RO5 | DF-DDFF | FB-DCFF |
| | RO11 | DF-DDFFFFFFFF | FB-DCFFFFFFFE |
| | RO20 | FD-DDFFFFFFFFFFDFFDFFD | FD-DCFFFFFFFFFFFDFFDFFD |
| Pattern-2 | RO5 | FE-FFFD | FE-FFFC |
| | RO11 | FE-FFFFFFFFFD | FE-EFFFFFFFFC |
| | RO20 | FD-FDFFDFFDFFFFFFFFFFD | FD-FDFFDFFDFFFFFFFFFFFC |
| Pattern-3 | RO5 | FC-FFFF | FC-FFFF |
| | RO11 | DF-FFFFFFFFDF | FB-FFFFFFFFDF |

### 3.3. Data Acquisition

During the experiment, the lowest temperature was $T_1$ = 25 °C, and the highest temperature was $T_2$ = 50 °C. A hot gun was used to increase the heat, and a cold gun was used to reduce the temperature. Two thermocouples were attached above and below the surfaces of the Cyclone V-chip. We considered the average of both surfaces' measurements as the chip's temperature. Figure 8 shows the data acquisition technique used in this work, i.e., a hardware configuration of a PC, Digilent Digital Discovery II, and an FPGA board. Digilent Digital Discovery II, a PC-based logic analyzer, was managed via WaveForms. Pattern Generator (run through WaveForms) was used to provide input configurations to trigger FPGA boards. The recorded pulses of all counters were viewed using Logic Analyzer (run through WaveForms) with a 500 kHz sample frequency.

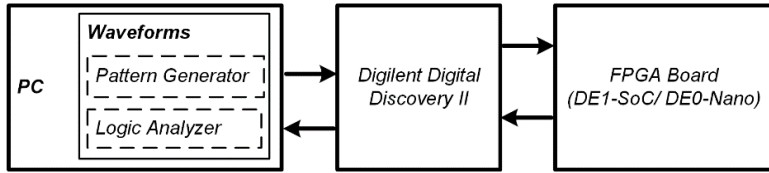

**Figure 8.** Data acquisition technique.

### 4. Performance Metrics

Over longer runtimes, the number of pulses resulted may exceed the counter capacity. We adjusted the recorded pulses before estimating the PUF metrics. We evaluate the performance in the coming sections, mainly based on uniqueness, reliability, uniformity, and bit aliasing. An analysis of those parameters is presented using several statistical parameters, such as standard deviation, frequency range, and frequency differences among ROs.

### 4.1. Estimating the Pulse Count

The pulses generated by RO may exceed the counter limits, thereby lowering the reliability [5]. These excess pulses mainly occurred when the runtimes were longer than 2 ms, 4 ms, and 8 ms in RO5, RO11, and RO20, respectively. Therefore, it was necessary to adjust the recorded pulses to estimate the actual number of pulses generated at each RO. For instance, Figure 9 shows the adjusted pulses versus the runtime of RO5 realization using Pattern-1 in chip-10. The adjustment process added the multiple of $2^{20}$ so that the pulses became linear with respect to runtimes. The number of pulses generated using the runtime of 4 ms was increased by $1 \times 2^{20}$. The number of pulses generated with a runtime of 6 ms was increased by $2 \times 2^{20}$. Likewise, with the pulses generated using a runtime of 8 ms, the amount was increased by $3 \times 2^{20}$.

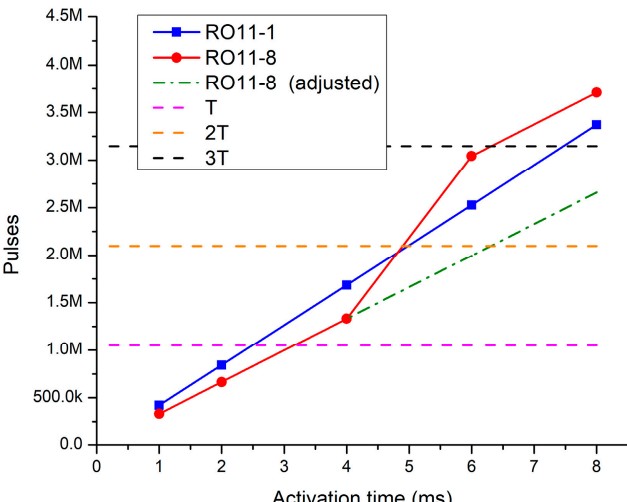

**Figure 9.** The plot of adjusted pulses versus runtimes of RO5 using Pattern-1 in chip-10.

As shown in Figure 9, a nonlinear trend (RO5-8) exists, shown in the red line starting at 4 ms, suggesting that pulses fell outside the green areas at 6 and 8 ms. In addition, this line indicates that pulses fell on the counting edge (a gray area), as shown in Figure 6. Although we chose the runtime selectively, the pulses might have approached the counting edge due to differences in delay among chips and the influence of temperature. Since the pulses generated by chain RO5-8 did not exceed $2 \times 2^{20}$ (at 6 ms) and $3 \times 2^{20}$ (at 8 ms), totals of pulses were increased by $1 \times 2^{20}$ (at 6 ms) and $2 \times 2^{20}$ (at 8 ms). Then the linear line was retrieved, as shown by the dash-dot line (green).

### 4.2. Uniqueness and Reliability

Uniqueness and reliability are crucial parameters for evaluating RO quality for PUF applications. Hamming distance (HD) is used to determine uniqueness based on differences among chip responses. For reliability, the calculation is performed by comparing the responses of internal chips. The uniqueness is computed using Equation (7) [13]. $c$ is the number of chips (board); $rb$ is the number of response bits; $u$ and $v$ are two boards being compared; $R_u$ and $R_v$ are the response bits with the equal runtimes generated by $u$ and $v$ boards. $HD(R_u, R_v)$ is the Hamming distance of response bits of c chips.

$$Uniqueness = \frac{2}{c(c-1)} \sum_{u=1}^{c-1} \sum_{v=u+1}^{c} \frac{HD(R_u, R_v)}{rb} \times 100\%, \qquad (7)$$

When the temperature rises, the pulses generated by ROs decrease, which automatically decreases the frequency. At some ROs, the reduction in pulses is higher than others, causing a different response when operated at higher temperatures. The difference in response is used as a reference to measure reliability using Equations (8) and (9) [34,35]. $k$ is the number of experiments on the same chip. $R_s$ is the response bit from chip $i$ at the lower temperature. Meanwhile, $R_{s,t}$ is the $t$-th sample of $R'_s$ response bit from chip $i$ at the higher temperature.

$$HD\ Intra = \frac{1}{k} \sum_{t=1}^{k} \frac{HD(R_s, R'_{s,t})}{rb} \times 100\%, \qquad (8)$$

$$Reliability = 100\% - HD\ Intra, \qquad (9)$$

Table 4 shows the uniqueness and reliability of all patterns in RO5, RO11, and RO20 realizations. The realization of RO5 using Pattern-3 delivered the closest uniqueness to the ideal value, and the realization using Pattern-1 produced the worst uniqueness,

40.52%. Meanwhile, the highest reliability occurred when ROs were realized using Pattern-1, 99.2%; and the lowest reliability when ROs were realized using Pattern-2, 97.96%. Like the situation of RO5, in RO11, the best uniqueness was produced using Pattern-3, and the worst uniqueness resulted from using Pattern-1—47.69% and 46.51%, respectively. While the realization using Pattern-3 produced the highest reliability, the realization using Pattern-1 delivered the lowest reliability. However, the differences in reliability in RO11 realizations using every pattern were less than 1%. In the realization of RO20, the one using Pattern-2 resulted in uniqueness and reliability slightly closer to the ideal compared to Pattern-1.

**Table 4.** Ring oscillator performance of all patterns and all stages.

| | Uniqueness (%) | | | Reliability (%) | | | Uniformity (%) | | | Bit aliasing (%) | | |
|---|---|---|---|---|---|---|---|---|---|---|---|---|
| | RO5 | RO11 | RO20 | RO5 | RO11 | RO20 | RO5 | RO11 | RO20 | RO5 | RO11 | RO20 |
| 1-13 Pattern-1 | 40.52 | 46.51 | 47.31 | 99.20 | 97.96 | 98.80 | 60.04 | 63.78 | 61.24 | 59.78 | 63.69 | 61.16 |
| Pattern-2 | 44.46 | 47.33 | 47.48 | 97.96 | 98.09 | 99.16 | 59.69 | 60.62 | 62.89 | 59.65 | 60.71 | 62.98 |
| Pattern-3 | 50.98 | 47.69 | - | 98.18 | 98.27 | - | 56.49 | 58.40 | - | 59.44 | 58.80 | - |

Highlighted values are closer to ideal.

### 4.3. Uniformity and Bit Aliasing

Uniformity and bit aliasing are used to measure the ratio between "1" and "0" in a series of bit responses. Uniformity determines the randomness of a specific chip's responses, and bit aliasing estimates the randomness of a particular bit's response in a group of chips. The ideal uniformity and bit aliasing values are 50%, which means the numbers of 1s and 0s in a series of response bits are equal. Uniformity is calculated based on the comparison of the Hamming weights (HW) of 1s on the same chip, according to Equation (10). $R_{s,l}$ is the $l$ bit from the sequence of response bits of a particular chip. Bit aliasing is measured based upon the Hamming weight of 1s among a group of chips using Equation (11). $R_{s,j}$ is the $j$ bit from the sequence of response bits of a group of chips.

$$Uniformity = \frac{1}{rb} \sum_{l=1}^{rb} R_{s,l} \times 100\%, \tag{10}$$

$$Bit - aliasing = \frac{1}{c} \sum_{j=1}^{c} R_{s,j} \times 100\%, \tag{11}$$

Table 4 shows the uniformity and bit aliasing of all patterns in RO5, RO11, and RO20. The uniformity and bit aliasing values in the table represent the percentages of 1s in the bit responses. In general, response bits contain more 1s than 0. In RO5, uniformity and bit aliasing were lower when ROs were realized using Pattern-3. Similarly, the realization of RO11 also produced no uniformity and bit aliasing close to ideal, especially when the ROs were realized using Pattern-1. Similarly, for RO20, both patterns produced more 1s than 0s. The realization of RO20 using the two patterns produced uniformity and bit aliasing above 60%.

### 5. Performance Analysis

The uniqueness, reliability, uniformity, and bit aliasing of ROs, to compare the performances, were presented in Section 4. The variations of performance among patterns were analyzed by assessing some statistical parameters of the pulses. The range of runtimes was calculated using Equations (1)–(6). The comparison of CRP models and our set-up is presented at the end of this section.

### 5.1. Statistical Properties

The chips of the DE1-SoC boards were numbered 1 to 4, and the chips of the DE0-Nano boards were numbered 5 to 10. Figure 10 shows the average frequencies of RO5, RO11, and RO20 in all chips for all patterns. As shown in the figures, chip-7 was the slowest, and chip-5 was the fastest. The differences in frequencies among patterns were due to various input configurations, as listed in Table 3.

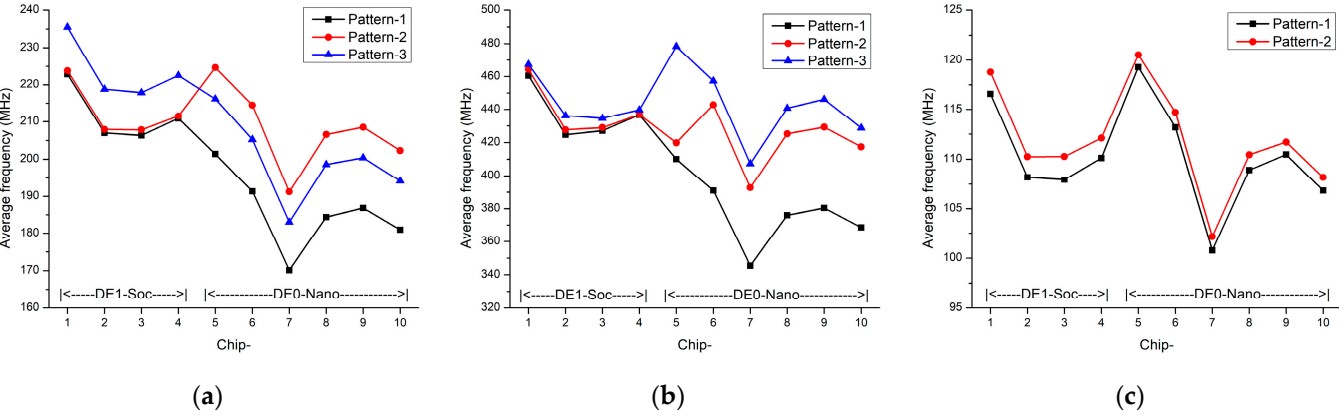

**Figure 10.** Average frequencies in all chips: (**a**) RO5, (**b**) RO11, (**c**) RO20.

In the RO5 realization, as shown in Figure 10a, the average frequency varied from around 345 MHz to 478 MHz. The realization of RO5 using Pattern-3 produced more pulses than others, especially in DE0-Nano boards, which is reflected by the input configuration of FC-FFFF, as listed in Table 3. Figure 10b shows the average frequency of RO11. In DE1-SoC, based on the configuration of LUTs, as listed in Table 3, the average frequency of ROs using Pattern-2 should have been higher than that of those using Pattern-3. However, Pattern-3 produced a higher average frequency because routing hotspots of ROs using this pattern (43.7%) were lower than those of Pattern-2 (51%), as listed in Table 2. Figure 10c shows the average frequencies of Pattern-1 and Pattern-2 in the RO20 realizations. The average frequency of RO20 varied from 100 to 120 MHz, about half of RO11's range. Thus, there were slight frequency differences between the two patterns for all chips.

Figure 11 shows the frequency ranges and standard deviations of all chips. For RO5, there existed significant differences in frequency ranges among patterns on DE0-Nano. The differences in frequency were due to average routing hotspots and input LUTs, as listed in Tables 2 and 3. The plots explained why Pattern-1 resulted in the least uniqueness, as listed in Table 4: high standard deviation and the high frequency ranges of some chips, as shown in Figure 11a. For RO11, there were minor differences in routing hotspots among patterns, as listed in Table 2. Therefore, the frequency ranges and standard deviations among patterns were similar, as shown in Figure 11b, which explains the slight uniqueness difference, as listed in Table 4. In RO20, the slight differences in frequency range and standard deviation between patterns, as shown in Figure 11c, explains the minor performance differences, as listed in Table 4.

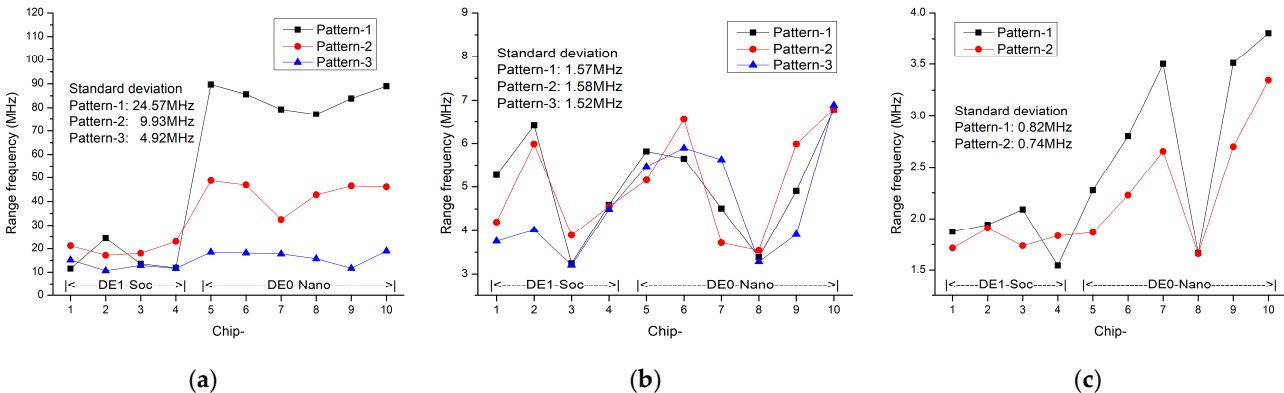

**Figure 11.** Frequency range and standard deviation in all chips: (**a**) RO5, (**b**) RO11, (**c**) RO20.

### 5.2. Runtimes Analysis

Merli et al. stated that the uniqueness was becoming stagnant after 204.8 μs [5]. This study found a similar behavior, where there were no significant changes in uniqueness and other metrics upon increasing the runtime. We analyzed the runtimes of RO11 and RO20 implemented using Pattern-3 and Pattern-2, respectively. The selection was made due to closer uniqueness to the ideal than other patterns, as shown in Table 4.

Whenever two or more ROs generate equal pulses, bit flipping happens when the ROs are run in different environmental conditions. For use as a PUF, all ROs should produce different pulses that are scattered from each other. In previous work [36,37], any RO pairs with a frequency difference less than the set threshold have been ignored, reducing the number of response bits. Instead of ignoring pairs, in this study we increased the runtime until RO produced a minimum pulse difference $s$. The runtime estimates were based on the minimum frequency differences among ROs. The minimum frequency differences of RO11 and RO20 in this work were 62.55 and 36.81 kHz, respectively. Hence, for s pulses, the shortest runtimes of RO11 and RO20 were ($s$/62.55) ms and ($s$/36.81) ms. Upon that realization, $s$ was chosen to be significantly higher. For instance, with $s$ = 100, the shortest runtimes of RO11 and RO20 were 1.598 and 2.717 ms, respectively.

Figure 12 shows the runtime predictions of RO11 (Pattern-3) and RO20 (Pattern-2) calculated using Equations (1)–(6). For RO11, as shown in Figure 12a, we suggest running ROs for $t_{\_H1}$ = 1.598 to 4.30 ms ($\Delta P_{\_H1}$ = 2.77%); $t_{\_L2}$ = 6.12 ms ($\Delta P_{\_L2}$ = 3.53%) to $t_{\_H2}$ = 8.61 ms ($\Delta P_{\_H2}$ = 5.54%); $t_{\_L3}$ = 12.24 ms ($\Delta P_{\_L3}$ = 7.06%) to $t_{\_H3}$ = 12.91 ms ($\Delta P_{\_H3}$ = 8.31%). Meanwhile, for RO20, as shown in Figure 12b, this work suggests running ROs for about $t_{\_H1}$ = 2.717 to 8.37 ms ($\Delta P_{\_H1}$ = 3.09%); $t_{\_L2}$ = 10.97 ms ($\Delta P_{\_L2}$ = 3.81%) to $t_{\_H2}$ = 16.74 ms ($\Delta P_{\_H2}$ = 6.19%); $t_{\_L3}$ = 21.93 ms ($\Delta P_{\_L3}$ = 7.62%) to $t_{\_H3}$ = 25.10 ms ($\Delta P_{\_H3}$ = 9.28%). Whenever ROs are activated at extreme temperatures (0 °C or 85 °C), the maximum pulses change with respect to counter capacity at a specific runtime is $\Delta P$.

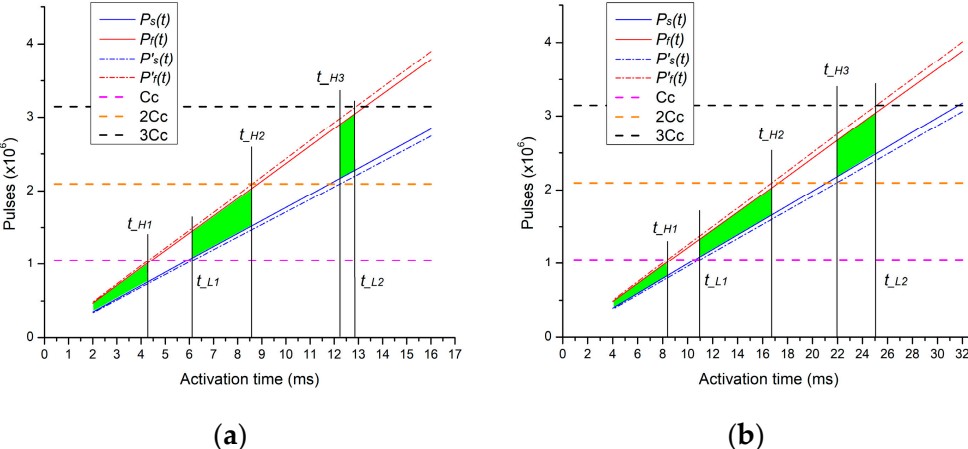

**Figure 12.** Prediction of pulses (**a**) RO11 using Pattern-3; (**b**) RO20 using Pattern-2.

*5.3. CRP and Metric Comparisons*

Table 5 shows the comparisons of the number of CRPs to the number of ROs. Our set-up can generate more responses due to more CRPs compared to other works, without a CRP enhancement technique (CRP inside an FPGA chip). In comparison, for $n = 16$ ROs, the CRPs of this work numbered 120, which is by far the highest compared to other models. Hence, our set-up requires a lesser area for realizing Ros, which allows more space for primary circuits. The CRPs of our set-up may also be realized outside the FPGA chip.

**Table 5.** Comparison of the numbers of CRPs.

| Author | No. of CRPs | CRP Enhance/Location |
|---|---|---|
| Suh et al. [3] | $n/8$ | no/inside chip |
| Maiti et.al. [31] | $n - 1$ | no/inside chip |
| Merli et.al. [38] | $n/2$ | no/inside chip |
| Yin et.al. [39] | $\log_2 n!$ | no/inside chip |
| Our set-up | $n!\,/2(n - 2)!$ | yes/inside or outside |
| Maiti et.al. [22] | $2^n - n - 1$ | yes/outside chip |
| Delavar et.al. [23] | $2^n - 1$ | yes/outside chip |

$n$ = number of ROs.

Furthermore, we compared the performances of the proposed uniform RO-PUF with other works in terms of uniqueness, reliability, uniformity, and bit aliasing. Most of the proposed RO-PUF that can be found in the literature utilize Xilinx FPGA, but some work is realized on Altera FPGA. Table 6 lists the metrics of the published works compared to those from this work. The authors improved RO quality in several previously published works using different techniques [5,7,14,22]. Most of the previously proposed works are inapplicable due to massive area usage for creating ROs, CRP generation, or the controlling circuit. We found that the ideas proposed by Maiti et al. [22] and Delavar et al. [23] are applicable due to CRP enhancing techniques. Both designs are better than this work in terms of uniqueness, but have lesser reliability scores, as listed in Table 6. Moreover, upon realization, Delavar et al. [23] compared the design quality using five chips only, implying that the ideal uniqueness is higher than 50% [13].

Chauhan et al. [40] proposed a different frequency characterization to improve reliability based on Delavar et al.'s idea [23]. However, since no expanded CRP techniques were used during response generation, it required a considerable area. Recently, Deng et al. proposed modifying the available configurable ring oscillator (C-RO) using hybrid logic gates [41]. The idea was realized on a 28 nm Xilinx chip with the help of 90 nm Spartan 3 for extracting reliability upon voltage variation. Upon comparing RO-PUFs, this study did not classify or identify the aging impact that may shift the frequency [42–44]. For instance,



high temperatures set up and its duration for extracting data from chips may degrade the frequency.

On a similar FPGA platform (Altera, San Jose, United States), in 2012, Bernard et al. implemented the ROs using Cyclone II and Cyclone III [20], making them the first in the literature to realize ROs as PUF on Altera. However, there were no metrics mentioned in the paper. Later then, Sahoo et al. [45] also realized the RO-PUF on Cyclone III. However, the reliability was measured at room temperature, and the design resulted in the worst bit aliasing. Finally, Feiten et al. [14] realized the RO-PUF using Cyclone IV. They described in detail the realization of Altera FPGA, but they were unable to improve the uniqueness. Meanwhile, this study was the first to realize RO using Cyclone V (28 nm). We found that our method is the best among those who realized their designs on Altera devices.

**Table 6.** A comparison of the metrics of the proposed RO-PUF with the metrics of others.

| | Uniqueness (%) | Reliability (%) | Uniformity (%) | Bit-Aliasing (%) | Platform |
|---|---|---|---|---|---|
| Suh et al. [3], 2007 | 46.15 | 99.52 | - | - | Xilinx (90 nm) |
| Maiti et al. [31], 2009 | 35.91–45.90 | - | - | - | Xilinx (90 nm) |
| Merli et al. [5], 2010 | 43.40–48.51 | 99.20, 98.28 | - | - | Xilinx (90 nm) |
| Xin et al. [46], 2011 | 32, 41 | 99.29 | - | - | Xilinx (90 nm) |
| Maiti et al. [22], 2012 | 49.99–50.07 | $\pm$92 *, $\pm$70 * (45 °C) | 50.02, 49.4 | 50.02, 49.4 | Xilinx (90 nm) |
| Feiten et al. [14], 2013 | 6.68 *–37.03 * | 99.41 *–82.5 * | 50.00 *, 62.07 * | - | Altera (65 nm) |
| Sahoo et al. [45], 2013 | 47.57 | 90.70 *** | 47 | 14.95 | Altera (65 nm) |
| Kodytek et al. [7], 2016 | 48.42–48.74 | 98.22, 97.55 | - | - | Xilinx (90 nm) |
| Delavar et al. [23], 2016 | 49.81 | 96.07 | - | - | Xilinx (90 nm) |
| Chauhan et al. [40], 2019 | 49.9 | 97.85–99.80 | - | - | Xilinx (28 nm) |
| Deng et al. [41], 2020 | 49.95 | 91.4–99.13 * | 49.61 | - | Xilinx (28 nm) |
| This work ** | 47.48 | 99.16 | 62.89 | 62.98 | Altera (28 nm) |
| Ideal value | 50% | 100% | 50% | 50% | |

* Approximate value; ** Pattern-2 (RO20); *** reliability was measured at room temperature.

For implementation in Cyclone V, we required 354 of the total available 15,880 ALMs (2% using DE0-Nano) or 30,070 ALMs (1% using DE1-SoC). The design also utilized 270 registers. However, most of the resources were used by control circuits. Upon realization of a fixed runtime, resource usage may be reduced. This work requires $10 \times 20$ ALMs to create RO20 or $10 \times 11$ ALMs to create RO11 and a number of (200 ALMs + 200 registers) to realize counters. The design would approximately draw 315 mW (DE0-Nano) or 411 mW (DE1-SoC) of static power, estimated using the Power Analyzer Tool (available in Quartus).

## 6. Conclusions

Analysis of the runtime of the proposed area-efficient uniform RO-PUF on the Cyclone V-chips has been carried out. The direct connection of ROs to counters of our set-up resulted in more CRPs than others with CRPs realized inside the FPGA chips. Hence, we allowed more space for the primary circuit. For implementation, the routing hotspots are designed to be uniform by rearranging logic gates associated with the respective ROs. This work suggests a careful selection of RO runtimes to avoid further bit flips with temperature variation. The shortest runtime was evaluated based on the frequency difference, which guaranteed the minimum difference of pulses among ROs is at least 100. Therefore, the 11-stage RO should run from 1.598 to 4.30 ms, from 6.12 to 8.61 ms, or from 12.24 to 12.91 ms. Meanwhile, the 20-stage RO should run from 2.717 to 8.37 ms, from 10.97 to 16.74 ms, or from 21.93 to 25.10 ms. In general, the performance of this set-up was better than the previously proposed methods for a similar platform (Altera), and the reliability is as good as that of the latest works using the same IC technology (28 nm). Moreover, the reliability of this is superior to that of the RO-PUF with CRP enhancements. Hence, when designing the RO-PUF for balancing area utilization and performance, the study strongly recommends using uniform routing hotspots and selective runtimes.



**Author Contributions:** Conceptualization, N.S. and S.F.W.M.H.; formal analysis, N.S., S.F.W.M.H. and M.S.A.T.; investigation, Z.Z.; methodology, Z.Z., N.S. and S.F.W.M.H.; resources, S.F.W.M.H.; software, Z.Z. and M.S.A.T.; supervision, N.S.; validation, Z.Z., S.F.W.M.H. and M.S.A.T.; visualization, M.S.A.T.; writing—original draft, Z.Z.; writing—review and editing, Z.Z., N.S., S.F.W.M.H. and M.S.A.T. All authors have read and agreed to the published version of the manuscript.

**Funding:** This research received no external funding.

**Conflicts of Interest:** The authors declare no conflict of interest.

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
