# Peer review of "Runtime Analysis of Area-Efficient Uniform RO-PUF for Uniqueness and Reliability Balancing"

_electronics, doi:10.3390/electronics10202504_

Round 1
Reviewer 1 Report
This paper presents a uniform RO-PUF with high reliability and uniqueness. This paper is well organized, and the design method is described clearly. Besides, sufficient experiments have been attached to support the research. Some comments following may help to improve this paper
1) In this paper, the authors adopt even number buffers and a modified NAND to implement the ROs. What`s the difference between this implementation and some references with even number inverters instead of buffers.
2) System noise, such as jitter and potential locking, may help to improve the randomness and unpredictability of the PUF. Some proper use of these signals in the design can help to improve robustness.
3)Using well-recognized test suits, such as NIST (A Statistical Test Suite for Random and Pseudorandom Number Generators for Cryptographic Applications), may provide a more persuasive voice for the design performance.
4)Some minor spell or grammar checks are required. For example, in Line70, "produce" should be "produced".
Reviewer 2 Report
The authors present a novel realization of ring oscillator physical unclonable functions (RO-PUF) requiring less area than conventional design. The proposed realization aims to balance the reliability and uniqueness of the RO-PUF. The authors have realized three different patterns of RO with three different lengths (9 different realizations), done performance analysis and compared the proposed RO-PUF with other state-of-the-art realizations.
The topic of the paper is within the scope of the journal. The paper is well structured and prepared; I recommend accepting the paper.
Minor Comments:
- The last two sentences of Abstract are a little bit confusing. It is unclear what the construction "is about 1.598ms to 4.30ms, or 21 6.12ms to 8.61ms, or 12.24ms to 12.91ms." means. It should be better specified where is the difference between "1.598ms to 4.30ms”, “21 6.12ms to 8.61ms", etc.
Typos:
- Please, use space between values and units according to the typography rules, e.g. https://physics.nist.gov/cuu/Units/checklist.html (line 18 “28 nm” and so on)
- line 67: please, specify the meaning of CRP before the first use of the abbreviation
- table 6, headline: there is a typo “Uniq.ueness”
- line 491: it seems that “Please add” should not be here
